# MicroRNA Expression Profile Changes after Cardiopulmonary Bypass and Ischemia/Reperfusion-Injury in a Porcine Model of Cardioplegic Arrest

**DOI:** 10.3390/diagnostics10040240

**Published:** 2020-04-21

**Authors:** Attila Kiss, Stefan Heber, Anne-Margarethe Kramer, Matthias Hackl, Susanna Skalicky, Seth Hallström, Bruno K. Podesser, David Santer

**Affiliations:** 1Ludwig Boltzmann Institute for Cardiovascular Research at Center for Biomedical Research, Medical University of Vienna, 1090 Vienna, Austria; attila.kiss@meduniwien.ac.at (A.K.); anne-margarethe.kramer@meduniwien.ac.at (A.-M.K.); david.santer@usb.ch (D.S.); 2Institute of Physiology, Center for Physiology and Pharmacology, Medical University of Vienna, 1090 Vienna, Austria; stefan_heber@icloud.com; 3TAmiRNA GmbH, 1110 Vienna, Austria; matthias.hackl@tamirna.com (M.H.); susanna.skalicky@tamirna.com (S.S.); 4Division of Physiological Chemistry, Otto Loewi Research Center, Medical University of Graz, 8010 Graz, Austria; seth.hallstroem@medunigraz.at; 5Department of Cardiac Surgery, University Hospital Basel, 4031 Basel, Switzerland

**Keywords:** microRNA, ischemia/reperfusion, cardiopulmonary bypass, cardioprotection

## Abstract

Identification of microRNAs (miRNA) associated with cardiopulmonary bypass, cardiac arrest and subsequent myocardial ischemia/reperfusion may unravel novel therapeutic targets and biomarkers. The primary aim of the present study was to investigate the effects of cardiopulmonary bypass and temperature of cardioplegic arrest on myocardial miRNA profile in pigs’ left ventricular tissue. We employed next-generation sequencing to analyse miRNA profiles in the following groups: (1) hearts were arrested with antegrade warm St Thomas Hospital No. 2 (STH2) cardioplegia (*n* = 5; STH2-warm, 37 °C) and (2) cold STH2 (*n* = 6; STH2-cold, 4 °C) cardioplegia. Sixty min of ischemia was followed by 60 min of on-pump reperfusion with an additional 90 min of off-pump reperfusion. In addition, two groups without cardiac arrest (off-pump and on-pump group; *n* = 3, respectively) served as additional controls. STH2-warm and STH2-cold cardioplegia revealed no hemodynamic differences. In contrast, coronary venous creatine kinase-myocardial band (CK-MB) levels were significantly lower in pigs receiving STH2-warm cardioplegia (*p* < 0.05). Principal component analysis revealed that cardiopulmonary bypass and cardioplegic arrest markedly affected miRNAs in left ventricular tissue. Accordingly, ssc-miR-122, ssc-miR-10a-5p, ssc-miR-193a-3p, ssc-miR-499-3p, ssc-miR-374a-5p, ssc-miR-345-5p, ssc-miR-142-3p, ssc-miR-424-5p, ssc-miR-545-3p, ssc-miR-30b-5p, ssc-miR-145-5p, ssc-miR-374b-5p and ssc-miR-139-3p were differently regulated by cardiopulmonary bypass (false discovery rate (FDR) < 0.05 versus off-pump group). However, only ssc-miR-451 was differently expressed between STH2-warm and STH2-cold (FDR < 0.05). These data demonstrate for the first time that cardiopulmonary bypass and temperature of cardioplegic solution affected the expression of miRNAs in left ventricular tissue. In conclusion, specific miRNAs are potential therapeutic targets for limiting ischemia-reperfusion injury in patients undergoing cardiac surgery.

## 1. Introduction

In cardiac surgery, the number of elderly and multimorbid patients has dramatically increased over the last 20 years, which overall results in increased peri- and post-operative mortality [1,2]. Therefore, constant efforts are needed to enhance intraoperative myocardial protection and, subsequently, improve postoperative outcomes. Ischemia and the subsequent reperfusion injury (IR) are triggered by marked calcium overload and oxidative stress of cardiomyocytes, ultimately causing myocardial apoptosis and inflammation. Cardiac arrest is routinely induced by cardioplegic solutions to allow coronary artery bypass surgery (CABG), cardiac transplantation and aortic valve procedures. In general, myocardial protection is ideally achieved by using warm or cold cardioplegic solution. However, the debate about the right application temperature is still ongoing. In 2004, in the United Kingdom, 14% of cardiac surgeons used warm and 56% cold blood cardioplegia for on-pump CABG [3]. Accordingly, cold blood might increase the affinity of haemoglobin towards oxygen and, thus, limit oxygen delivery [4]; furthermore, the prospective randomised warm heart trial showed reduced creatine kinase-myocardial band (CK-MB) release as well as less postoperative low cardiac output after warm cardioplegia. However, short- [5] and long-term mortality were comparable [6] in warm versus cold cardioplegia. Similarly, in our own hands in more than 2200 patients, there was no difference in 30-day mortality between warm and cold cardioplegia in adult cardiac surgery. The only group of patients that showed a significant benefit of warm cardioplegia were patients undergoing urgent revascularisation [7].

This study intends to elucidate microRNAs’ (miRNA) expression profile unique to cardiopulmonary bypass and cardiac arrest (warm and cold cardioplegia), which may contribute to cardioprotection or cardiac (dys-)functions. MiRNAs are endogenous non-coding RNA molecules which negatively regulate gene expression by inducing degradation of messenger ribonucleic acid (mRNA) or inhibiting translation. Both circulating and tissue miRNAs have potential value in diagnosis and prognosis of cardiovascular disease, including ST-elevation myocardial infarction, atherosclerosis, diabetes and heart failure [8,9,10]. A number of studies also demonstrate that miRNA may serve as a fundamental target to limit cardiac ischemia-reperfusion (IR)-injury in both acute myocardial infarction as well as during cardiac surgery. However, the impact of cardiopulmonary bypass (CPB) and cardiac arrest (cold and warm) on miRNA expression in association with functional outcome are unknown. Recently, we have investigated for the first time that the expression profile of miRNAs in left ventricular (LV) tissue following IR in a porcine model of CPB [11] and miRNA expression could be affected by cardiac arrest and myocardial IR. This follow-up study aimed to investigate the effect of CPB and temperature of cardiac arrest (cold at 4 °C and warm at 37 °C) on miRNA expression profile and whether this is associated with cardiac function in a porcine model of cardioplegic arrest. Of importance, we investigated for the first time whether miRNA expression within the LV was affected by CPB and cardioplegic arrest. Accordingly, we found that CPB significantly affected miRNA expression within the myocardium. Interestingly, the miRNAs affected by CPB are known to be cardioprotective. Moreover, only miR-451 expression was markedly higher in myocardium after warm cardioplegic arrest in association with less myocardial damage. These data concerning miRNA expression profiles may contribute to a better understanding of the mechanism of IR in the setting of CPB and cardiac arrest. Targeting miRNAs may serve as a novel therapeutic approach to limit IR-injury in patients undergoing cardiac surgery.

## 2. Materials and Methods

### 2.1. Animals

Female pigs (*n* = 18, Austrian Landrace) were housed at the Centre for Biomedical Research, Medical University of Vienna, Austria. Pigs arrived one week prior to experiments for acclimatisation, and were fed with a standard diet twice a day (ssniff GmbH, Soest, Germany) and water ad libitum. The experiments were approved by the Animal Ethics Committee of the Medical University of Vienna and the Austrian Ministry of Science and Technology (GZ: 66.009/0171-II/3b/2011). All animals received humane care in compliance with the Federation of European Laboratory Animal Science Associations (FELASA). One experiment had to be stopped after opening of the pericardium due to a persisting ventricular arrhythmia; therefore, 17 pigs were included into the study.

### 2.2. Experimental Protocol

For premedication, Ketamine (15 mg/kg i.m.), Acepromazine (1.3 mg/kg) and Atropine (0.5 mg) were used. Animals were anaesthetised (2.5 mg/kg Propofol, 15 mg Piritramid and 20 mg Rocuroniumbromid), intubated and ventilated. Anaesthesia was maintained with Piritramid (1.875 mg/kg/h), Rocuroniumbromid (3.7 mg/kg/h) and Propofol (10 mg/kg/h). Pressure monitoring was performed via the left femoral artery, and a central venous catheter was placed into the left external jugular vein for venous access. After sternotomy, pigs were heparinised (300 IU/kg), ascending aorta (Optisite arterial cannula, Edwards Lifesciences Corp, Irvine, CA, USA) and the right atrium (Trim Flex venous cannula, Edwards Lifesciences Corp, Irvine, CA, USA) were cannulated and pigs were put on normothermic CPB (Stockert SIII Heart Lung System, Sorin/LivaNova PLC, London, UK). Pigs were randomised into the two study groups (STH2-warm, STH2-cold). Additional monitoring included a Swan-Ganz oximetry thermodilution catheter (741HF75P, Edwards Lifesciences Corp, Irvine, CA, USA) for cardiac output (CO), ultrasound flow probe for coronary flow (CF) of the left anterior descending coronary artery (LAD), a pressure tip catheter for left ventricular pressure (LVP, Millar Instruments, Houston, TX, USA) and a coronary sinus catheter to sample coronary effluent during the first 60 min of IR.

After baseline measurements, CPB was initiated, the aorta was cross-clamped and hearts were arrested with antegrade, blood-based warm (*n* = 5; STH2-warm) or cold STH2 (*n* = 6; STH2-cold). Sixty min of ischemia were followed by 60 min of on-pump reperfusion (sampling time points: 1, 5, 15, 30, 60 min; Figure 1), and another 90 min of off-pump reperfusion (time points: 90, 120 and 150 min; Figure 1). After venous and arterial decannulation, protamine (300 IU/kg) was administered. To maintain systolic blood pressure above 70 mmHg and haemoglobin above 6 mg/dl, both volume substitution and continuous noradrenaline infusion were administered. Samples from the left anterior wall were harvested for determination of miRNAs. The pigs were sacrificed with high-dose pentobarbital. Two additional groups served as controls to show that miRNA expression was primarily affected by CPB: In the off-pump group (*n* = 3), after sternotomy, pigs were heparinised (300 IU/kg) and after 210 min, sacrificed as described above. In the on-pump group (*n* = 3), no aortic cross-clamping was performed and, consequently, instead of ischemia (60 min), on-pump reperfusion was prolonged for 60 min (Figure 1).

### 2.3. Cardioplegic Solution

Cardioplegic solution (St Thomas No. 2 solution; NaCl: 110.0 mmol/L, NaHCO_3_: 10.0 mmol/L, KCl: 16.0 mmol/L, MgCl_2_: 16.0 mmol/L; CaCl_2_: 1.2 mmol/L) was administered in the two study groups, STH2-warm and STH2-cold. Crystalloid solution was provided by the hospital pharmacy of the General Hospital Linz, Austria, and was mixed (1:2; total volume: 1500 mL) with 500 mL pig blood immediately before administration. After aortic cross-clamping, 1000 mL of the respective blood cardioplegia was infused with a pressure of 60 mmHg and a temperature of 37 °C (STH2-warm) or 4 °C (STH2-cold). After 30 min of ischemia, an additional 500 mL of the cardioplegic solutions were infused. The cardioplegic solution consisted of 10.7 mmol/L of magnesium, 110 mmol/L sodium chloride, 10.7 mmol/L potassium chloride and 1.2 mmol/L calcium chloride.

### 2.4. Hemodynamic Evaluation

Heart rate (HR) and arterial pressure (AP) were recorded in all groups. In study groups (STH2-warm and STH2-cold), right atrial pressure (RAP), left ventricular pressure (LVP) and cardiac output (CO) were continuously measured, as well as coronary flow (CF) in the left anterior descending coronary artery (LAD). Wedge pressure were recorded at baseline (before start of CPB) and at 1, 5, 15, 30, 60, 90, 120 and 150 min of reperfusion. Echocardiographic evaluation—ejection fraction (EF)—was performed at baseline and before the sacrification. External heart work (EHW) was calculated by CO x systolic LVP for each time point.

### 2.5. Biochemical Analyses

Arterial blood samples were drawn at baseline, 1, 5, 15, 30, 60, 90, 120 and 150 min of reperfusion (R). During CPB, venous samples were drawn from the coronary sinus during controlled on-pump reperfusion: baseline, 1, 5, 15, 30 and 60 min of reperfusion. Immunoassays were performed for CK-MB (Cobas immunoassay CKL, ID 0-324, Roche, Germany). Lactate was analysed with blood gas measurements (ABL 800 flex, Drott, Austria).

### 2.6. Total RNA Extraction

Samples from the left anterior wall were stored at −80 °C prior to miRNAs analyses. Total RNA was extracted from homogenised heart tissue using the miRNeasy purification kit (Qiagen, Germany) and was quality checked for RNA integrity using the RNA 6000 bioanalyser assay (Agilent, CA) and spectrophotometric RNA quantification (Nanodrop), as described previously [11].

### 2.7. Small RNA Sequencing

Equal amounts of total RNA (250 ng) were used for small RNA library preparation using the Clean-Tag ligation kit (Trilink, CA). Adapter-ligated libraries were amplified using barcoded Illumina reverse primers in combination with the Illumina forward primer. Libraries were pooled at equimolar rates on the basis of a DNA-1000 bioanalyzer run (Agilent, CA) and sent for sequencing to Exiqon (Denmark). The small RNA library pool was sequenced on an Illumina NextSeq 500 with 50 bp cycle length. Reads were adapter-trimmed and filtered for low-quality reads (Phred Score (Q) < 30). MicroRNA annotation was performed on the basis of sequence alignments against the genome reference and miRBase release 21. Read counts were normalised to the total number of reads detected per sample to obtain the “tags per million” (TPM) for each miRNA and sample. Exploratory analysis was performed using ClustVis and differential expression analysis was performed using the EdgeR package under R/Bioconductor [12].

### 2.8. Reverse-Transcription Quantitative PCR (RT-qPCR) Messenger RNA Analysis

Analysis for the assessment of pro-inflammatory cytokines (TNF-alpha, IL-6 and HMGB1) and apoptosis (Caspase-3, Caspase-1, BCl-2)-related genes (list of primers sequences in Appendix A) was performed by RT-qPCR. Extracted and intact (RIN > 7) total RNA was used for cDNA synthesis using the TATAA GrandScript cDNA Synthesis Kit (TATAA Biocenter, Goteborg, Sweden). Reverse transcription reactions were performed in 20 µL reactions with 500 ng of total RNA. For each sample, qPCR reactions were performed in 10 µL reactions in duplicates with 2 µL 1:2 diluted cDNA and 8 µL qPCR Mix, which consists of 5 µL TATAA SYBR® Grandmaster Mix (TATAA Biocenter), 0.8 µL forward and reverse primer (10 µM) and 2.2 µL nuclease-free water. qPCR was performed on a Roche LightCycler 480 II instrument. The following thermocycling conditions were use: 30 s at 95 °C, 45 cycles of 5 s at 95 °C, 15 s at 63 °C, 10 s at 72 °C, followed by melting curve analysis. Cq values were computed using the second derivative maximum method provided with the LC480 II software. ACTB was use as a reference gene and normalisation was performed as follows: normalised Cq = Cq ACTB–Cq gene of interest (GOI). (Appendix A).

### 2.9. Statistical Analysis

Graphs were generated by GraphPad Prism 8.2 using analyses performed in IBM SPSS Statistics 26 (IBM Corp, Armonk, NY, USA). For the main manuscript, the pooled mean with 95% confidence interval is shown for the baseline measurement, as this was a randomised study and baseline differences thus occurred exclusively by chance. For all other time points, estimated arithmetic (in case untransformed data were analysed) or geometric (analyses of log-transformed data due to right-skewed distribution) are shown with 95% confidence intervals. Due to the baseline value covariate, these can be interpreted as the means occurring if all animals had exactly the same baseline values. In addition, the estimated group differences in the original unit (untransformed data) or as fold-change (log-transformed data) are plotted with 95% confidence intervals pooled for all time points (interaction not significant) or for each time point separately (significant interaction). In case of a non-significant interaction, group differences pooled over all time points are plotted, otherwise (significant interaction), group differences are shown for each time point separately. Due to the multiple dependent variables (miRNAs) and the exploratory character of this study, no formal sample size calculation was performed. Sample size was chosen pragmatically with regards to one of our previous studies. Results need to be interpreted accordingly. Specifically, non-significant group differences must not be interpreted as evidence for the null hypothesis. Statistical analysis: To estimate whether group differences changed over time, a mixed model approach was applied. First, approximate normal distribution was checked visually. Some data appeared markedly right-skewed and were log-transformed before analysis. The factor time point (with levels corresponding to the times of measurement of each dependent variable) was specified as a within-subjects factor, and the factor group (STH2-warm, STH2-cold, on-pump control and off-pump control) as between-subjects factors. In addition, the values of the respective dependent variable at time point 0 was included as a covariate in the model to adjust for baseline differences. All models were estimated using the restricted maximum likelihood (REML) method. An adequate covariance structure was chosen based on the smallest Akaike information criterion. In case of a significant interaction between time point and group, contrasts were used to estimate group differences at each time point. In case of a non-significant interaction, the term was dropped from the model and the group differences were estimated pooled for all time points. All reported *p*-values are the result of two-sided tests. *p*-values of 0.05 or less were considered significant. No adjustment for multiplicity was performed due to the exploratory character of this study. Confidence intervals need to be interpreted accordingly.

## 3. Results

### 3.1. Animal Characteristics

Seventeen pigs were included in the study (STH2-warm: *n* = 5, body weight (BW) 64 ± 5 kg, heart weight (HW) 322 ± 19 g, ejection fraction (EF) 58% ± 8%. STH2-cold: *n* = 6, BW 69 ± 10 kg, HW 294 ± 21 g, EF 63% ± 4%. Off-pump: *n* = 3, BW 64 ± 7 kg, HW 282 ± 33 g, EF 57% ± 11%. On-pump: *n* = 3, BW 73 ± 5 kg HW 308 ± 32 g, EF 69% ± 0%).

### 3.2. Hemodynamic Data

After adjustment for baseline differences, there was no evidence that during reperfusion the differences between STH2-warm and STH2-cold changed over time regarding systolic LVP (timepoint x group interaction *p* = 0.93, Figure 2A) and cardiac output (CO, *p* = 0.98, Figure 2B). Additionally, systolic AP (timepoint x group interaction *p* = 0.24,), mean arterial pressure (*p* = 0.58,), diastolic arterial pressure (*p* = 0.97,), HR (*p* = 0.41,), RAP (*p* = 0.84,), stroke volume (SV, *p* = 0.15,), external heart work (EHW, *p* = 0.57,) and noradrenaline demand (*p* = 0.75,) were also comparable between the two groups. After baseline adjustment, the change over time of coronary flow during reperfusion was significantly different between the STH2-warm and STH2-cold cardioplegia groups (timepoint x group *p* < 0.001, Figure 3A). After 15 min of reperfusion, pigs receiving STH2-warm showed a 0.41-fold altered coronary flow compared to the STH2-cold group (95% CI 0.20–0.89). After 30 min, a similar reduction was found (0.35-fold; 95% CI 0.17–0.76). After this time point, CF values did not differ significantly between study groups (Figure 3B). In addition, after 15 min, pigs receiving STH2-warm also exhibited 0.42-fold (95% CI 0.18–0.97) lower CF values when compared to the off-pump control (Figure 3B). Left ventricular ejection fraction was similar between the groups at the end of experiments (STH2-warm: 61% ± 7%, STH2-cold: 71% ± 11%, on-pump: 69% ± 0%).

### 3.3. Biochemical Data

Arterial CK-MB levels were generally lower (0.82-fold, 95% CI 0.67–1.00, *p* = 0.05) in pigs receiving warm STH2 cardioplegia compared to cold STH2 cardioplegia, whereby the difference in arterial CK-MB levels were independent of the time course (Figure 4A). In line with this, CK-MB levels measured in the coronary venous blood were also generally lower in pigs receiving warm STH2 (0.76-fold, 95% CI 0.68–0.84, *p* = 0.0004, Figure 4B). Arterial lactate concentration showed significant differences between groups (timepoint x group interaction *p* = 0.004, data not shown) with a significant reduction in STH2-warm at timepoint 150 min of reperfusion compared to STH2-cold. There was no difference in arterial lactate between on-pump and off-pump groups. The time course of the coronary lactate concentration was similar between groups (timepoint x group interaction *p* = 0.08, data not shown). However, pigs receiving warm or cold STH2 cardioplegia generally had significantly higher levels than pigs in the off-pump control group (data not shown).

### 3.4. Expression of MicroRNAs in Left Ventricular Tissue Samples

Overall, more than 300 miRNAs were detected in all samples with a minimum read count of 5 TPM. Principal component analysis (Figure 5A) showed that cardiopulmonary bypass and cardiac arrest resulted in a specific microRNA expression profile (Figure 5B). Accordingly, the following miRNA were differently expressed between off-pump versus on-pump: ssc-miR-122, ssc-miR-10a-5p, ssc-miR-193a-3p, ssc-miR-499-3p, ssc-miR-374a-5p, ssc-miR-345-5p, ssc-miR-142-3p, ssc-miR-424-5p, ssc-miR-545-3p, ssc-miR-30b-5p, ssc-miR-145-5p, ssc-miR-374b-5p and ssc-miR-139-3p (false discovery rate (FDR) < 0.05, Figure 6 and Appendix A). Of note, only ssc-miR-451 was differently expressed between STH2-warm and STH2-cold (Figure 7 and Appendix A).

### 3.5. Expression of Pro-Inflammatory Cytokines and Apoptosis Markers

Systemic inflammation and myocardial apoptosis are common effects of CPB. Therefore, we assessed the mRNA expression levels of pro-inflammatory cytokines (*Tnf-α* and *Hmgb1*) and apoptosis-related genes (*Casp3*, *Casp1* and *Bcl2*) within the myocardium (Figure 8A–F). The expression of *Tnf-α* was unaffected by CPB. In contrast, in both STH2-warm and STH2-cold cardioplegia, an increased expression of *Tnf-α* compared to the off-pump group was observed (Figure 8A, *p* < 0.01, respectively). Similarly, the expression of IL-6 tangentially increased in STH2-warm and STH2-cold cardioplegia compared to the off-pump group (Figure 8B). It is noteworthy that *Hmgb1* was markedly upregulated in STH2-cold cardioplegia versus the off-pump group (Figure 8C, *p* < 0.01). Expression of apoptosis-related markers, *Casp1* and *Bcl2* mRNA, was comparable between the groups (Figure 8D,F). In contrast, upregulation of *Casp3* expression was observed in cardiac arrest (STH2-cold and STH2-warm) groups compared to both the off-pump and the on-pump groups (Figure 8E, *p* < 0.01, respectively). However, there was no difference in expression between STH2-cold and STH2-warm cardioplegia.

## 4. Discussion

Despite the achievements in cardiac surgery, CPB and cardioplegic arrest are key triggers of myocardial injury during cardiac surgery. This is because the heart is exposed to global ischemia and, therefore, susceptible to reperfusion injury. This study characterised the effects of warm and cold blood STH2 cardioplegia during prolonged on-pump and off-pump reperfusion after 60 min of cardiac arrest on miRNAs expression, hemodynamic variables, expression of inflammatory cytokines and apoptosis-related genes, as well as markers of cardiac injury. Although haemodynamic recovery was similar between groups, significantly lower CK-MB levels after warm compared to cold STH2 cardioplegia indicate reduced myocardial damage with warm cardioplegia. It is accepted knowledge that blood cardioplegia is safe and might provide superior cardioprotection (higher oxygen content, improved buffering capacity, oxygen extraction and preserved high-energy phosphates) in comparison with crystalloid cardioplegia [13]. Nevertheless, the discussion about the ideal temperature of blood cardioplegia is still ongoing. Hypothermic cardioplegic arrest prevents ischemic damage by downregulation of myocardial metabolism, while warm blood cardioplegia offers metabolic as well as haemodynamic improvements [14,15], which is widely acknowledged by the application of a final cardioplegic perfusion (“hot shot”) before aortic declamping [13]. In this study, we found that warm cardioplegic arrest provided better haemodynamic stability. Furthermore, in our model, we observed reduced release of coronary venous CK-MB in STH2-warm during reperfusion.

To the best of our knowledge, for the first time, we found evidence that CPB and STH2 cardioplegia indeed leads to induction of a specific microRNA expression profile within the left ventricle. Accordingly, CPB also significantly affected miRNAs’ expression within the myocardium. Interestingly, the miRNAs which are affected by cardiopulmonary bypass are known to initiate cardioprotective mechanisms. Previous studies demonstrated that miR-1, miR-499 and miR-208 are associated with and contribute to the pathophysiology mechanisms of IR-injury in patients with acute myocardial infarction [16,17]. Nevertheless, there are only few studies that have investigated the changes of miRNA profiles during cardiac surgery. Accordingly, the elevation of miR-1, mir-133 and miR-499 levels in plasma and atrial tissue are associated with myocardial damage in open-heart surgery with CPB [18,19,20]. Additionally, aberrant miRNA expression may implicate cardiac dysfunction, and thus, make them potential biomarkers and therapeutic targets during elective cardiac surgery. Nevertheless, there is no comparative study that characterised miRNA expression (left ventricular tissue) in CPB and after cardioplegic arrest. With next-generation sequencing, more than 300 miRNAs (with average TPM > 5) were identified in left ventricular tissue. First, we performed principal component analysis between all study groups in order to identify whether the experimental groups (STH2-warm and STH2-cold) were the major determinant of miRNA expression in cardiac tissue, or whether there might be other confounding factors that induce miRNA expression. It is noteworthy that principal component analysis showed a clear difference between off-pump and on-pump groups, suggesting that miRNA expression was crucially affected by CPB. In our study, we compared the expression of microRNA after CPB and found a significant reduction in ssc-miR-122, ssc-miR-10a-5p, ssc-miR-193a-3p and increase in ssc-miR-499-3p, ssc-miR-374a-5p, ssc-miR-345-5p, ssc-miR-142-3p, ssc-miR-424-5p, ssc-miR-545-3p, ssc-miR-30b-5p, ssc-miR-145-5p, ssc-miR-374b-5p and ssc-miR-139-3p expression compared to the on-pump group. Among the downregulated miRNA, miR-122 plays a pathophysiological role in IR-injury by the promotion of apoptosis [21,22]. Interestingly, the expression of miR-122 in cardiac tissue is very low. Nevertheless, previous studies have demonstrated its pathological role in various cardiovascular diseases [23,24]. This suggests that myocardial miR-122 might serve as a novel therapeutic target to reduce apoptosis and, subsequently, improve cardiac recovery in patients undergoing CABG. In addition, upregulation of miR-499-3p, miR-142-3p and miR-145-3p initiate cardioprotection via activation of anti-apoptotic and anti-inflammatory pathways in cardiomyocytes [25,26,27]. An important finding of our study is the upregulation of miR-142-3p in cardiac tissue in the on-pump group. MiR-142-3p is a repressor of components of the NF-kB pathway [26,28] and is upregulated in bridge-to-transplant candidates with a left ventricular assist device (LVAD) and a consecutive higher cardiac index compared to controls without LVAD [29]. In line with this, our data demonstrate that mechanical circulatory support (on-pump group) markedly increases levels of cardioprotective miRNA.

Next, we found that induction of cardiac arrest with STH2 cardioplegia results in a specific miRNA expression profile: ssc-miR-451 was significantly decreased in STH2-cold compared to STH2-warm cardioplegia. Interestingly, miR-451 and miR-144 are considered to play a role in remote ischemic conditioning (RIC)-mediated cardioprotection [30]. In our study, ssc-miR-451 was significantly increased and there was a tendency for ssc-miR-144 upregulation in the STH2-warm group, which was in association with the reduction in CK-MB levels. More recently, RIPHeart [31] and the ERICCA [32] clinical trials demonstrated that RIC did not improve clinical outcomes in patients undergoing elective on-pump CABG with or without concomitant valve surgery. In addition, follow-up studies tried to clarify why remote ischemic preconditioning failed to improve protection in patients undergoing cardiac surgery. Interestingly, we found that CPB itself and warm STH2 cardioplegia markedly changed the expression of cardioprotective microRNAs. Of note, RIPHeart and the ERICCA trials mainly administered cold cardioplegia [31,32]. Therefore, our results, particularly the upregulation of cardioprotective miRNA and dysregulation of miR-451 and miR-144, may provide further explanations for the inefficiency of RIC in cardiac surgery. However, further studies are needed for clarification. 

Development of the CPB revolutionised cardiac surgery and contributed immensely to improved patient outcomes, although it is associated with the activation of different coagulation and pro-inflammatory signalling (systemic circulation) pathways. In this context, another interesting finding of the current study were the effects of CPB on the myocardial expression of inflammatory and apoptotic markers [33]. There is abundant evidence that CPB induces damage of myocardial and vascular endothelial cells [34,35]. Moreover, the biological hallmark of myocardial injury following cardioplegic arrest is strongly associated with the increase of apoptosis and inflammation in cardiac tissue and subsequently, initiates temporary cardiac dysfunction. Accordingly, we also found that cardioplegic arrest and IR markedly increase the mRNA expression of *Caspase-3* but not *Caspase-1* in comparison to the off-pump group. More recently, it has been described that NLRP3 (NOD-, LRR- and pyrin domain-containing protein 3) inflammasome is triggered by locally released damage-associated molecular patterns (DAMPs) and that NLRP3 amplifies the inflammatory response, cell death through Caspase-1 activation and the inhibition of Caspase-1 reduced IR-injury in acute MI [36]. In contrast to this finding, our study demonstrates that Caspase-1 upregulation does not play a dominant role concerning apoptosis in CPB. However, one can speculate that there are different mechanisms of apoptosis in acute MI and CPB. Further studies are needed to clarify the different pathways. The expression of anti-apoptotic *Bcl-2* was also not affected by CPB. Besides apoptosis, the deleterious role of inflammation in cardiac surgery has been demonstrated in previous preclinical and clinical studies [34]. Accordingly, the overexpression of HMGB1 initiates the upregulation and release of pro-inflammatory cytokines, such as TNF-α, IL-6 and IL-1β, which subsequently contribute to early and late cardiac functional impairment in the setting of CPB [37]. Consistently, we observed an upregulation of *TNF-α* and IL-6 showed a tendency to increase in both STH2 groups compared to the off-pump group. Nevertheless, these changes certainly contribute to impaired cardiac recovery independently of the temperature of the cardioplegic solution.

In summary, our study, for the first time, demonstrates that CPB markedly enhanced specific miRNAs in LV tissue, which are known to be cardioprotective. In addition, use of warm cardioplegia resulted in a marked decline in coronary CK-MB levels compared to cold cardioplegia. This effect was associated with enhanced miR-451 levels in the myocardium. In conclusion, these data suggest that miRNAs are a possible target for reducing IR-injury in patients undergoing cardiac surgery.

## 5. Clinical Perspectives


(1)microRNAs play a major role in myocardial ischemia and reperfusion injury and targeting microRNAs may be a potential therapeutic approach to achieve cardioprotecion in patients with acute myocardial infarction or elective cardiac surgery.(2)To the best of our knowledge, for the first time, we found evidence that CPB and cardioplegic arrest followed by IR leads to a specific microRNA expression profile change within the left ventricle in a translational large animal model. Notably, CPB-affected miRNAs’ are known to initiate cardioprotection.(3)Targeting specific miRNAs may be a potential therapeutic target for limiting IR injury in patients undergoing cardiac surgery.


## 6. Study Limitations

Several limitations of the present study have to be considered. First, experiments were performed on healthy, female, young pigs to model cardiopulmonary bypass and cardioplegic arrest, while the typical patient undergoing elective cardiac surgery is older and has comorbidities and risk factors that affect the outcome of any therapy. Animals underwent on-pump reperfusion for 60 min to guarantee controlled reperfusion under comparable flow rates, pressures and haemoglobin. Analysis of cardiac enzymes from coronary sinus blood was performed under these standardised conditions. Although a sample size was defined, it is possible that between-group differences might be caused by a sampling error due to the low number of pigs. In addition, this study does not provide pressure–volume relationships since measurements were performed with a pressure-tip catheter. Since correction for multiple testing has not been performed, the miRNA findings in this study are of an exploratory nature and require further replication to verify their relevance for the regulation of gene activity in hearts undergoing cardioplegic arrest.

## Figures and Tables

**Figure 1 diagnostics-10-00240-f001:**
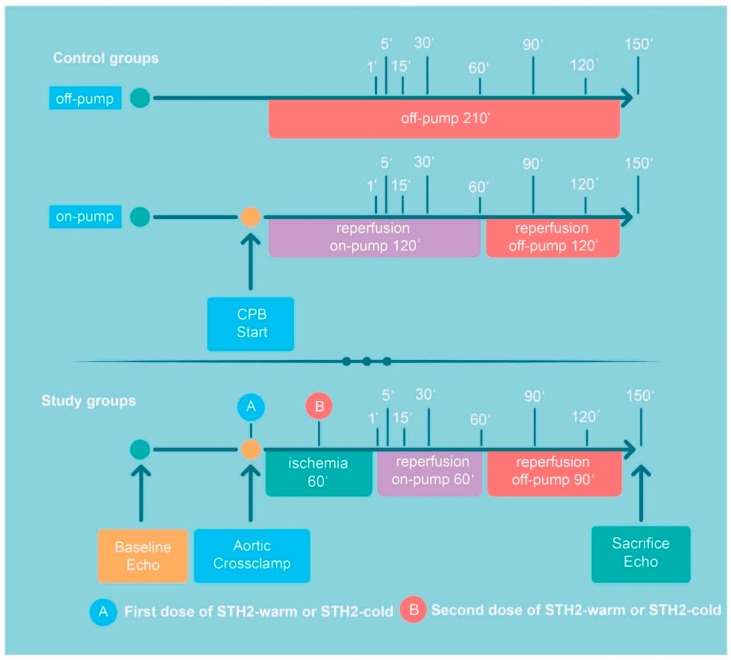
Timeline of the experimental protocol. After baseline haemodynamic and echocardiography measurements pigs were randomised to the following groups: (1) in the off-pump group, after sternotomy, pigs were heparinised (300 IU/kg) and after 210 min, sacrificed. (2) In the on-pump group, no aortic cross-clamping was performed and, consequently, instead of ischemia (60 min), on-pump reperfusion was prolonged for 60 min. In the STH2-warm group and STH2-cold group, aortic cross-clamping was performed followed by 60 min of ischemia, 60 min of on-pump and, finally, 90 min of off-pump reperfusion. Before sacrifice, echocardiography was repeated. A indicates administration of the first dose of STH2-warm or STH2-cold cardioplegic solution (1000 mL after aortic cross-clamping), B indicates administration of the second dose of STH2-warm cardioplegic solution or STH2-cold cardioplegic solution during ischemia (500 mL). The indicated times refer to the sampling points. CPB: cardiopulmonary bypass and STH2: St Thomas Hospital No. 2.

**Figure 2 diagnostics-10-00240-f002:**
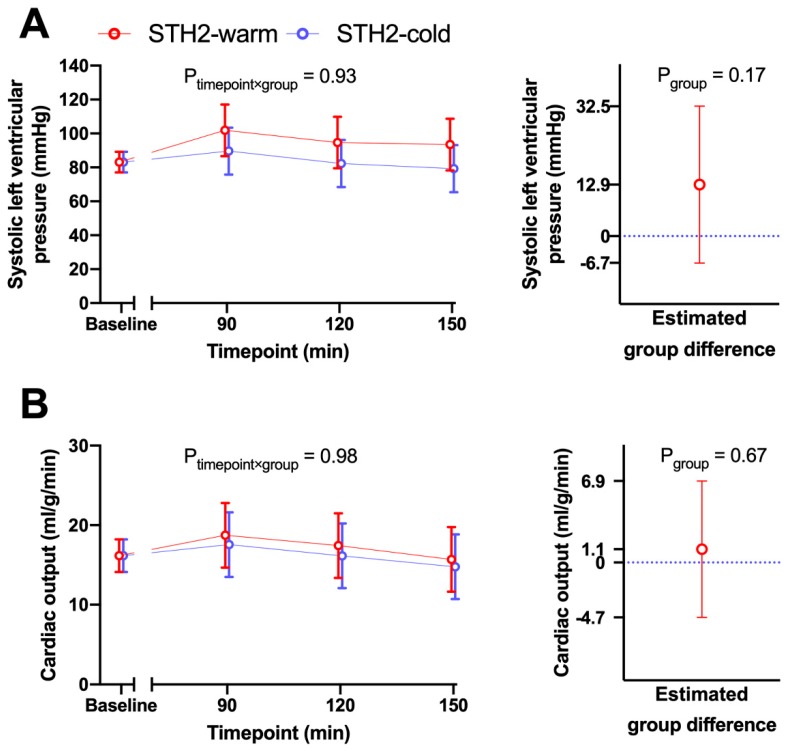
Left ventricular pressure and cardiac output. Haemodynamic data were recorded at baseline and every 30 min during off-pump reperfusion. Systolic left ventricular pressure (**A**) and cardiac output (**B**) were comparable in STH2-warm versus STH2-cold. Values are least square arithmetic means (left) or estimated mean group differences with 95% confidence intervals, *n* = 5–6/group.

**Figure 3 diagnostics-10-00240-f003:**
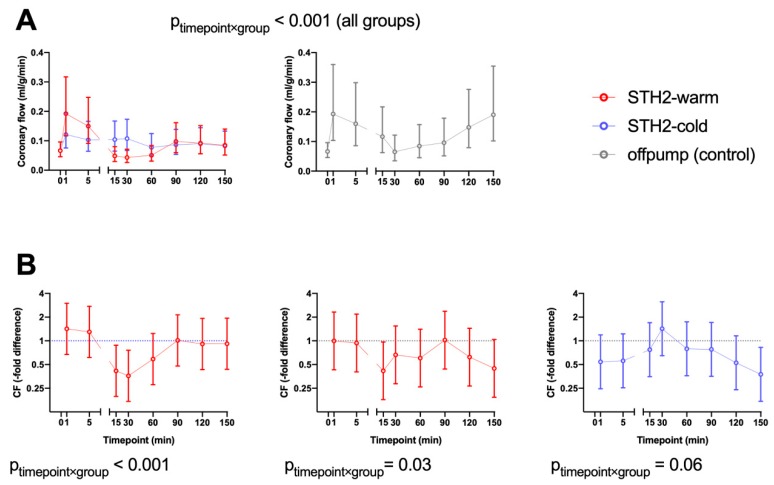
Changes of coronary flow (CF) during reperfusion. After baseline adjustment, the change over time of coronary flow (**A**) during reperfusion was differently affected by STH2-warm cardioplegic solution and STH2-cold cardioplegic solution (timepoint x group *p* < 0.001). After 15 min of reperfusion, pigs receiving STH2-warm showed a 0.41-fold altered coronary flow compared to the STH2-cold group (95% CI 0.20–0.89). After 30 min, a similar reduction was observed (0.35-fold; 95% CI 0.17–0.76). At further timepoints, CF values did not significantly differ between groups. After 15 min, pigs receiving STH2-warm also exhibited 0.42-fold (95% CI 0.18–0.97) lower CF values compared to the off-pump control (**B**). Values are estimated geometric means (**A**) or estimated -fold differences (**B**) with 95% confidence intervals, *n* = 5–6/group. CF: coronary flow.

**Figure 4 diagnostics-10-00240-f004:**
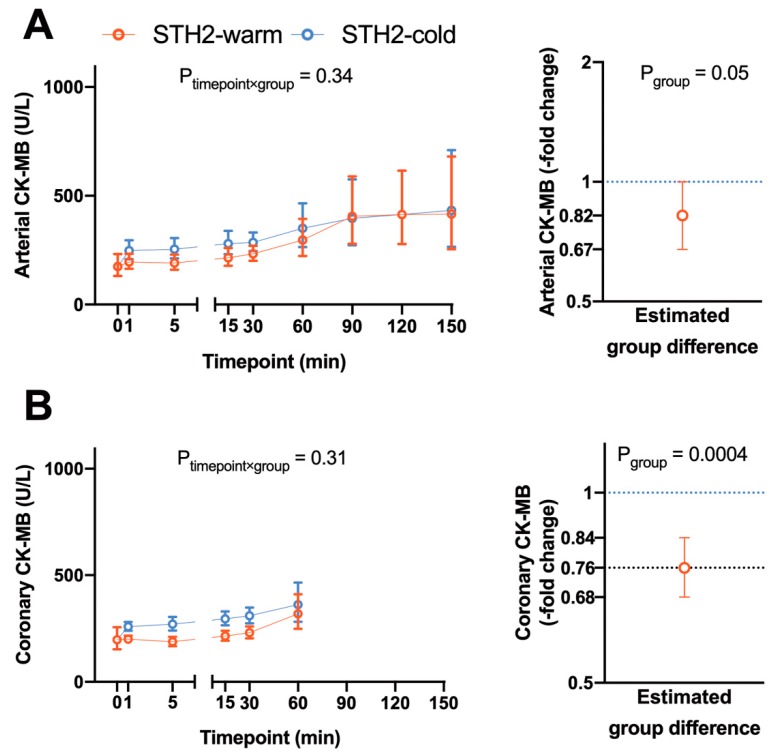
Levels of circulating creatine kinase during reperfusion. Arterial CK-MB (**A**) levels were generally lower (0.82-fold, 95% CI 0.67–1.00, *p* = 0.05) in pigs receiving warm STH2 cardioplegia compared to the cold STH2 group, whereby the differences in arterial CK-MB levels were independent of the time course. CK-MB levels (**B**) measured in the coronary venous blood were also generally lower in pigs receiving warm STH2 cardioplegia (0.76-fold, 95% CI 0.68–0.84, *p* = 0.0004). Values are estimated geometric means (left) or estimated -fold differences (right) with 95% confidence intervals, *n* = 5–6/group. CK-MB: creatine kinase-myocardial band.

**Figure 5 diagnostics-10-00240-f005:**
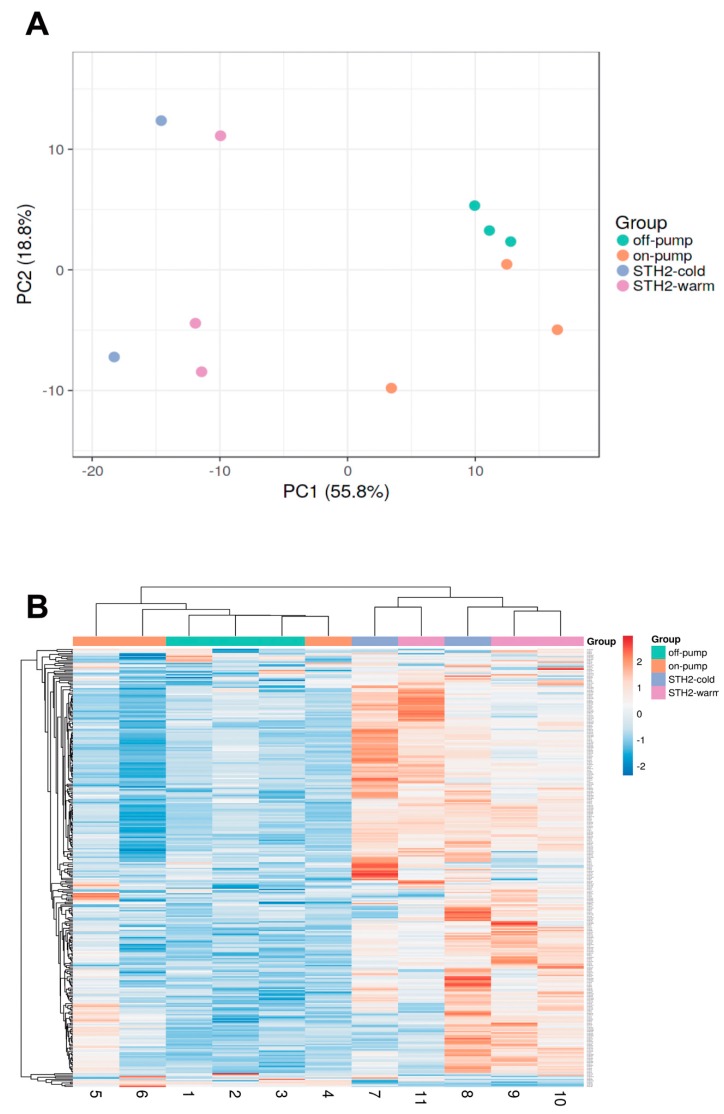
Principle component analysis on microRNA expression. (**A**) Principal component analysis on microRNA expression (*n* = 212, TPM) > 1) within left ventricular tissue of pigs from on-pump, off-pump, STH2-warm and STH2-cold groups. (**B**) Heat map and cluster analysis (Euclidean distance) of 212 commonly expressed miRNAs. *n* = 3–6/groups. PC: principal component. TPM: tags per million

**Figure 6 diagnostics-10-00240-f006:**
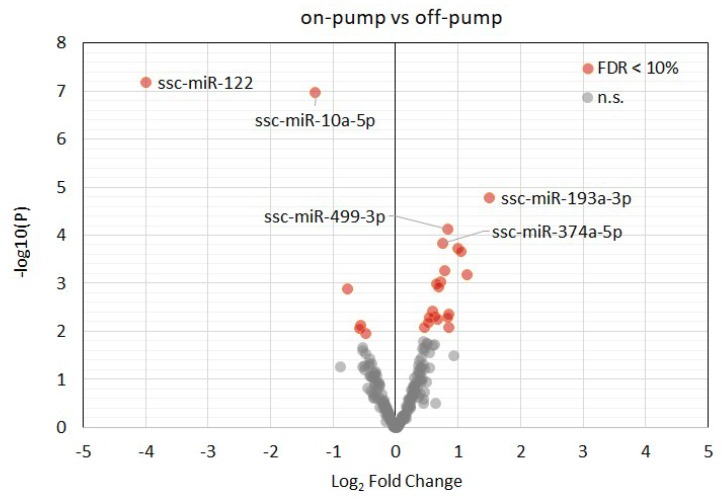
Volcano plot of the detected myocardial miRNAs in pigs with off-pump compared with on-pump protocols. Upregulated and downregulated miRNAs are in red. Top 5 statistically significant (adjusted *p*-value (FDR) < 0.1) miRNAs are depicted with their miRBase identifiers. FDR: false discovery rate.

**Figure 7 diagnostics-10-00240-f007:**
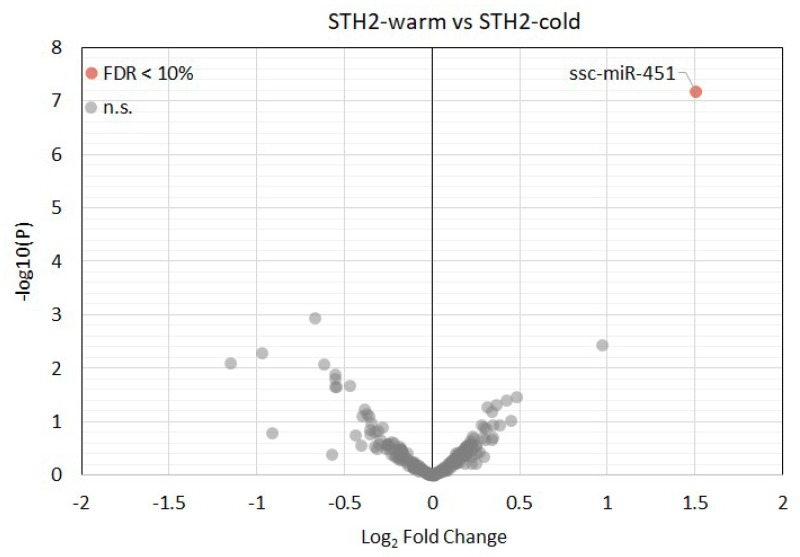
Volcano plot of the detected myocardial miRNAs in pigs with STH2-warm and STH2-cold cardioplegia arrest. Upregulated and downregulated miRNAs are in red. Statistically significant (adjusted *p*-value (FDR) < 0.1) miRNAs are depicted with their miRBase identifiers. FDR: false discovery rate.

**Figure 8 diagnostics-10-00240-f008:**
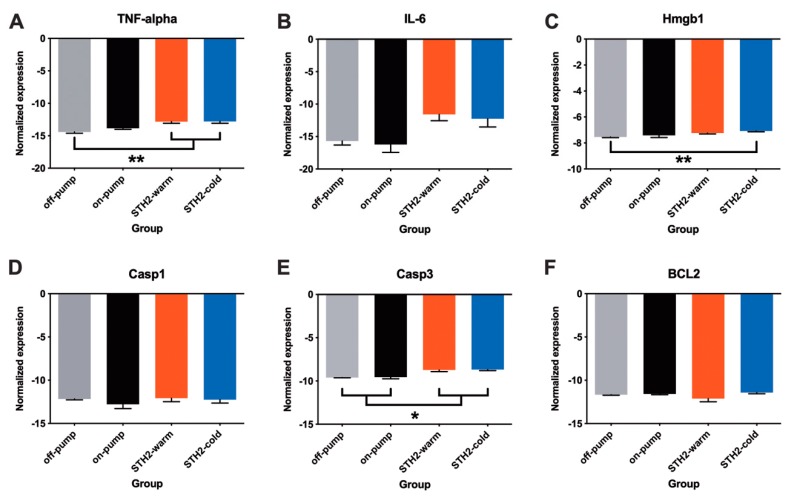
Expression of pro-inflammatory cytokines and apoptotic markers within the myocardium. (**A**) mRNA expression of Tnf-α, (**B**) IL-6, (**C**) Hmgb1, (**D**) Casp3, (**E**) Casp1 and (**F**) Bcl2 within left ventricular tissue samples in all experimental groups. Values are mean ± SD, *n* = 5–6/group. IL-6: Interleukin-6, Tnf-α: Tumour Necrosis Factor alpha; Hmgb1: High mobility group box protein 1, Casp1: Caspase 1 and Casp3: Caspase 3. * = *p* < 0.05, ** = *p* < 0.01.

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
