# Peer review of "MicroRNA Expression Profile Changes after Cardiopulmonary Bypass and Ischemia/Reperfusion-Injury in a Porcine Model of Cardioplegic Arrest"

_diagnostics, 2020, doi:10.3390/diagnostics10040240_

Round 1

Reviewer 1 Report

The authors studied the effect of ischemia and reperfusion in a porcine model of CPB on LV myocardial miRNA expression and assessed the potential influence of cardioplegic solution temperature (37 C vs 4 C).  In this model 60 min of ischemia was induced by aortic cross clamping followed by 60 min on pump and 90 min off pump reperfusion. Coronary venous CK levels were reduced in the warm cardioplegia group.  The authors’ principal findings were that a large array of miRNA showed differential regulation in expression due to CPB whereas only one (miR-451) was affected by temperature. Because some mi-RNAs are likely involved in cardioprotection the authors propose that based on their findings, mi-RNAs may be therapeutic targets to attenuate ischemia-reperfusion injury in patients undergoing cardiac surgery.

COMMENTS

The authors have performed a study which involves the measurement of numerous miRNAs which may offer potentially interesting insights into their role in cardioprotection. Unfortunately, I found this paper to be rather descriptive and one which offers a paucity of specific details especially in terms of a cause and effect relationship.  Thus, while the authors present numerous miRNA changes there is no specific connection between these changes and protection and only hypothetical evidence is provided.

Other points:

  1. Why are there no differences in function?
  2. I am not sure what the take home message is with respects to cytokines or markers of apoptosis. There seems to be few differences here.
  3. Where are the before/after clamping echo data? I seem to have missed it.
  4. Why were only female pigs used? The ages of these pigs do not seem to have been provided.  Are they hormonally protected? If so, the results would be very difficult to interpret.     

Author Response

COMMENTS

The authors have performed a study which involves the measurement ofnumerous miRNAs which may offer potentially interesting insights into their role in cardioprotection. Unfortunately, I found this paper to be rather descriptive and one which offers a paucity of specific details especially in terms of a cause and effect relationship.  Thus, while theauthors present numerous miRNA changes there is no specific connection between these changes and protection and only hypothetical evidence isprovided.

We thank the reviewer for these comments. Indeed, our MS is rather descriptive, but the aim was to characterize for the first time the impact of cardiopulmonary bypass and cardiac arrest by cardioplegic solutions (warm and cold) on microRNAs expression in LV cardiac tissue by unbiased NGS approach. We did not aimed to target microRNAs and discover their causative role in myocardial damage/protection due to cardiopulmonary bypass and ischemia and reperfusion injury after cardioplegic arrest. However, future studies are needed to clarify whether the present microRNAs are play role in myocardial protection or injury during CPB surgery. This limitation has been acknowledged in the revised version of the MS.

    Other points:

    Why are there no differences in function?

We thank the reviewer for these valuable comments.  In previous studies and meta-analyses, no clear hemodynamic superiority has been reported for warm or cold blood cardioplegia (Abah et al. “Is cold or warm blood cardioplegia superior for myocardial protection?”, Interactive CardioVascular and Thoracic Surgery, 2012). Therefore, we were not surprised that -besides the differences in CF- hemodynamic function was comparable in STH2-warm and STH2-cold in this study. Since the main focus of this manuscript was the difference between warm and cold administration, we chose a solution that is (1) well established (2) clinically applicable and (3) that is successful in a warm as well as cold blood composition (Ibrahim et al. “A Clinical Comparative Study Between Crystalloid and Blood-Based St Thomas’ Hospital Cardioplegic Solution, Eur. J Cardiorthorac Surg, 1999).

I am not sure what the take home message is with respects to cytokines or markers of apoptosis. There seems to be few differenceshere.

We thank the reviewer for this comment. We assume our findings in respect to cytokines and apoptosis are relevant and important regarding reperfusion injury. Conceptually, cardioplegic solutions protect the heart against myocardial damage during elective cardiac surgery. However, reperfusion (after declamping) initiates reperfusion injury and certainly contributes to worse clinical outcome. Besides reperfusion injury, CPB is associated with the activation of different coagulation and pro-inflammatory signalling (systemic circulation). In this context, another important finding of the current study was the effects of CPB on the myocardial expression of inflammatory and apoptotic markers. Accordingly, we also found that cardioplegic arrest and IR markedly increase the mRNA expression of Caspase-3 but not Caspase-1 in comparison to the off-pump group. The expression of anti-apoptotic Bcl-2 was also not affected by CPB. In addition, the upregulation of HMGB1 initiates the release of pro-inflammatory cytokines, such as TNF-α, IL-6 and IL-1β, which subsequently contributes to early and late cardiac functional impairment in setting of CPB as described previously. Consistently, we observed an upregulation of TNF-α and IL-6 showed tendency to increase in both STH2 groups in compared to off-pump group. Therefore, targeting these cytokines may be an approach to reduce reperfusion injury in patients are subjected to CPB.

Where are the before/after clamping echo data? I seem to have missed it.

We thank the reviewer for this comment. These results have been included into the revised version of the MS (3.2 Hemodynamic Data Section. The baseline ejection fraction (before clamping) is noted in the “3.1 Animal Characteristics” section).

 Why were only female pigs used? The ages of these pigs do not seem   to have been provided.  Are they hormonally protected? If so, the results would be very difficult to interpret.

We thank the reviewer for this valuable comment. First, we have more experience to handle female pigs due to simple the size (male are much larger at same age). In addition, pigs are sexually mature at 5–7 months of age, our pigs are more juvenile and certainly not fully hormonally protected. Since there were no difference in age between the groups, only the treatments showed difference, therefore we assume this issue does not affect our results and conclusions. In addition, the pigs were randomized into the experimental groups. However, we acknowledged this limitation in the revised version of the MS.

Reviewer 2 Report

I would wish to congratulate authors on conducting a well-desgined and well-described study that offers some novel findings in term of cardioplegia and ischemia-reperfusion injury.

I have no major comments, except for the fact that authors did not discuss any potential limitations of their study. This should be included in a special paragraph within the Discussion section during the revision process.

Author Response

    I would wish to congratulate authors on conducting a well-desgined and well-described study that offers some novel findings in term of cardioplegia and ischemia-reperfusion injury. I have no major comments, except for the fact that authors did not discuss any potential limitations of their study. This should be  included in a special paragraph within the Discussion section during the  revision process.

The present study has the following limitations: Experiments were performed on healthy, female young pigs to model cardiopulmonary bypass and cardiioplegic arrest, while the typical patient subject to elective cardiac surgery (CABG and valve replacement) is older and presents with co-morbidities and risk factors that affect the response to any therapy.  In addition, animals underwent on-pump reperfusion of 60 min to guarantee controlled reperfusion under comparable flow rates, pressures and hemoglobin. Under these standardized conditions analysis of cardiac enzymes from coronary sinus blood was performed. Although a sample size was defined, it is possible that between-group differences might be caused by a sampling error due to the low number of pigs. In addition, this study does not provide pressure-volume relationships since measurements were performed with a pressure-tip catheter. Since correction for multiple testing has not been performed, the miRNA findings in this study are of exploratory nature and require further replication to verify their relevance for the regulation of gene activity in hearts undergoing cardioplegic arrest.  These limitation are acknowledged in the revised version of the MS.